# Trauma and resilience in an urban clinic for unhoused young adults: A mixed methods study

Shruti Arora[1]*, Allison Ong[2]*, Michael Wilkes[3], Hilary Aralis[4], Andres F. Sciolla[5]

1 University of California, Davis School of Medicine, Sacramento, California, United States of America, 2 University of Miami/Jackson Memorial Health, Miami, Florida, United States of America, 3 Office of Medical Education, Office of the Dean, and Department of Medicine, General Internal Medicine and Ethics, University of California, Davis School of Medicine, Sacramento, California, United States of America, 4 Department of Public Health Sciences, University of California, Davis, Sacramento, California, United States of America, 5 Department of Psychiatry and Behavioral Sciences, University of California, Davis, Sacramento, California, United States of America

* shrarora@ucdavis.edu (SA); allison.ong@jhsmiami.org (AO)

## Abstract

Unhoused youth and young adults report higher rates of adverse childhood experiences (ACEs) compared to the general population. ACEs are linked in a dose-dependent manner to poorer mental and physical health outcomes in adulthood. Resilience, which is the ability to manage stressors and recover from adversity, is a measurable quality with both intrinsic and extrinsic sources that can potentially be used to mitigate the negative effects of ACEs. This study aimed to explore the relationship between ACEs and resilience among unhoused transitional-aged youth (TAY) at an urban medical clinic. Our approach relied on qualitative thematic analysis of semi-structured interviews. The participants included twenty-eight unhoused patients aged 18–27 years attending a free medical clinic. Our main outcome measures focused on perceptions of trauma experienced by unhoused youth, ACEs score, PHQ-9, and Brief Resilience Scale (BRS) scores. The number of ACEs in our study population ranged from 6 to 15, with a mean of 9. The most frequently reported adverse childhood events were divorce, verbal abuse, substance use by someone in the household, feeling unloved or "unspecial," and physical abuse. Resilience scores in our population ranged from 9 to 30, with a mean of 17. Participants with high resilience were more likely to report close relationships, educational goals, connections with mental health professionals, and housing in transitional living programs (TLPs). Moving from research to policy, strategies that promote resilience by identifying intrinsic assets and external resources can be used to develop transformative programs for unhoused youth and young adults. Housing First initiatives are crucial for providing a safe and secure environment that helps youth feel capable of mitigating the impact of adversity on their health outcomes. Housing First initiatives are instrumental to providing a safe and secure environment for youth to feel capable of mitigating the impact of adversity on health outcomes.

**Data availability statement:** The minimal dataset of this study is in the paper and its Supporting Information files. The data we did not include are the GAD-7 survey results which only a few participants completed and were deemed irrelevant to the study, data on quantity of ACES experienced by each participant, and direct quotes that were later truncated or clarified for the manuscript. We felt that the data included in this manuscript were sufficient to the content of this paper. For legal and ethical reasons we do not want to release the full direct quotes of our participants who are unhoused transitional youth.

**Funding:** The author(s) received no specific funding for this work.

**Competing interests:** The authors have declared that no competing interests exist.

## Introduction

More than four million youth and young adults experience homelessness in the United States, highlighting the urgent need for targeted outreach and advocacy in this population [1,2]. Among these individuals, transitional aged youth (TAY)—those between the ages of 15 and 26—face heightened risks of suicidal ideation, food insecurity, substance use, and trauma [3,4].Approximately 40% of unhoused TAY identify as LGBTQ, a subgroup that experiences significantly higher rates of depression, post-traumatic stress disorder (PTSD), and suicide attempts compared to their heterosexual and cisgender peers [5–7].

Adverse childhood experiences (ACEs)—defined as traumatic events or conditions occurring before the age of 18—have been strongly linked, in a dose-dependent manner, to long-term physical and mental health consequences [8–13]. In children and adolescents, ACEs may contribute to developmental and educational delays, obesity, asthma, sleep issues, frequent infection, and higher levels of anxiety and depression [14,15]. This early exposure to toxic stress can contribute to a higher burden of chronic disease and premature mortality in adulthood [16,17].

Our research team was deeply engaged in understanding the complex interplay among childhood trauma, health outcomes, and housing instability. For over a decade, we provided free medical care at an urban youth center, which, until its funding ended in late 2023, served as a crucial access point for vulnerable TAY in our city. Although medical services have since ceased, the center continues to offer comprehensive wraparound support, including educational resources, legal advocacy, and housing referrals.

Throughout our clinical work, we observed that many TAY faced challenges extending well beyond their medical needs. As we supported them in setting goals and navigating adversity, we became increasingly interested in the role of resilience in shaping their trajectories. Resilience, the dynamic capacity to adapt to hardship and thrive with minimal psychological cost, is promoted by both internal *assets*, such as self-efficacy and self-esteem, and external *resources*, such as community programs and mentorship opportunities that promote coping skill development [18].

Higher levels of resilience have been associated with lower rates of depression and greater life satisfaction in adulthood [19,20]. Thus, resilience theory, as outlined by Fergus and Zimmermann, provides a compelling framework for exploring youth homelessness [18]. Our qualitative study aims to investigate the interrelationships among ACEs, resilience, and wellness outcomes to inform more responsive and holistic models of care for this population. Our findings offer foundational insights and hypothesis generation for future interventions, potentially serving as a guide for other clinics addressing the unique needs of unhoused youth. This work aims to fill existing gaps in both the literature and policy discussions, particularly around how to integrate trauma-informed, resilience-focused approaches into youth services.

## Methods

### Ethics statement

This mixed methods study was approved by the UC Davis institutional review board (IRB). Participants provided verbal consent and permission to use their

deidentified quotes for research purposes. We followed the Standards for Reporting Qualitative Research (SRQR) reporting guidelines [21].

We developed our interview questions based on existing literature on unhoused youth as well as our team's experiences working at the clinic. From January 2022 through August 2023 we invited TAY checking into the clinic or assessing other services at the youth center to participate. Participation in the interview did not affect their ability to receive health care. All participants provided informed verbal consent and were assigned unique de-identified codes to protect confidentiality. Each participant completed a series of validated instruments including the original ACEs questionnaire, a five-question Philadelphia Expanded ACEs Scale, a PHQ-9 if none was completed in the past six months, and the Brief Resilience Scale (Figures A-D in S1 Text). The Philadelphia Expanded ACEs scale builds upon the original by incorporating additional items related to neighborhood adversity and exposure to violence. The Brief Resilience Scale is a six-item tool that has been validated in over 140 studies, showing strong positive correlations with protective mental health factors and negative correlations with adverse outcomes [22].

Semi-structured interviews were conducted in a private setting to ensure participant comfort and confidentiality. Each interview included demographic questions such as gender and race/ethnicity identity. The interview guide was piloted with three TAY participants and feedback was used to revise questions for improved clarity and relevance, while still allowing space for new topics to emerge organically. With additional consent, interviews were audio-recorded for transcription and analysis.

Participants were included in the study if they had experienced homelessness at any time in their lives. Exclusion criteria included those who declined consent, did not speak English, did not complete all elements of the survey, or determination by the clinician that the individual was too ill or cognitively impaired to engage in a meaningful conversation.

## Qualitative analysis

We employed a thematic analysis approach to analyze the qualitative data. The analysis process followed these established steps:

1. Data familiarization: Researchers immersed themselves in the data by reading and re-reading interview transcripts.

2. Initial coding: Two researchers independently coded the transcripts, identifying key concepts and patterns in the data.

3. Theme development: Codes were grouped into potential themes, which were then reviewed and refined through team discussions.

4. Theme refinement: Themes were checked against the coded extracts and the entire dataset to ensure they accurately represented the data.

5. Theme definition: Clear definitions and labels were generated for each theme.

6. Report production: A final analysis was conducted, selecting compelling extract examples and relating the analysis back to the research question and literature.

The analysis was an iterative process, with the research team meeting regularly to discuss and refine the emerging themes until consensus was reached on the final thematic framework. We used Microsoft Excel to facilitate data management and coding.

## Quantitative analysis

Data was analyzed using SAS, version 9.4. Descriptive statistics including means, medians, standard deviations, frequencies, and percentages were generated for demographic variables and other measures among the sample as a whole. A two-level categorical variable was created to indicate TAY with high vs. low levels of reported ACEs. The median among

this sample was used to classify TAY into each of the two levels. An analogous approach was taken to create a two-level categorical variable for high vs. low resilience. Resilience scores above our mean of 17 were classified as "high", and scores 17 and below were classified as "low". Demographic and housing characteristics were compared across participants with high vs. low ACEs and across participants with high vs. low resilience using Pearson Chi-square and Fisher's Exact tests. Median age, resilience, ACEs and depression scores were compared across these classifications as well using nonparametric Mann-Whitney U Tests appropriate for small samples.

## Results

### Quantitative results

Thirty three TAY were interviewed; five were excluded because they did not meet inclusion criteria. Of the 28 TAY included in the analysis, 36% identified as male and 18% identified as female (Table 1). In regard to gender identity, 46% identified as LGBTQ, with most LGBTQ youth being transgender or non-binary. The top three categories for self-identified racial or ethnic identity were white (42%), mixed race (27%), and black (15%). Thirty-six percent of our youth reported being in foster care at some point in their lives. At the time of the interview, 50% of participants were staying at a shelter. Twenty-one percent lived on the streets or reported they were "in between" places to stay, for example, having been kicked out of a shelter the previous night or moving from one house to another. Eighteen percent were staying in transitional living facilities (TLFs) which allow residents longer stays while they are provided resources for career development, life skills, and transition to affordable housing.

ACEs scores in our sample ranged from 6 to 15 with a mean of 9.7 (standard deviation [SD] = 3.0). The top ACES were parental divorce/separation (experienced by 86%), verbal abuse (82%), substance use by someone in the household, (82%), feeling unloved or unspecial (78%), and physical abuse (64%). ACE endorsement varied by gender identity; for example, among female youth, verbal abuse was ranked first whereas among male youth it was ranked fourth. Among transgender youth, feeling unloved or unspecial was ranked first.

Resilience scores in our sample ranged from 9 to 30 with a mean of 17.4 (SD = 5.2). PHQ-9 scores ranged from 1 to 27 with a mean of 14.6 (SD = 7.0). Seventy-four percent of TAY reported PHQ-9 scores of 10 or higher indicating moderate-to-severe symptoms of depression. Table 2 displays differences with regards to ACES classification. PHQ-9 scores were significantly higher among TAY in the high ACES group (p = 0.0003). Resilience scores were higher in the high ACES group (median = 18.0) relative to the low ACES group (median = 16.0), although these differences were not statistically significant (p = 0.2110). Table 3 displays differences with regards to Resilience classification.

### Qualitative results

In the following sections, direct quotes from our participants explore three prominent themes of trauma and three themes of what we perceived as "resilience-promotive" factors that interviewees considered positive drivers in their life.

**Themes of trauma: Abuse.** Half of our sample reported emotional, physical, or sexual abuse as a child. Some were abused by nuclear family members including siblings and parents, particularly when a parent abused illicit substances. One youth was told by their parents that they were "a piece of shit," causing them to run away from home.

- *My leg was broken multiple times as a child and now I have issues walking. I get constant panic attacks, like I'm so scared that I don't eat for days. Then I binge-eat not knowing if I'm going to be able to eat again... When I was 17, I was kicked out of the truck we were living in and forced to walk in the cold – barefoot – down __ Blvd to stay with my aunt... she became verbally abusive and bullied me for money and rent. (Participant 17)*

- *I feel that sexual abuse has definitely affected my relationships. I feel detached from children and don't want to be around them. (Participant 13)*

**Table 1. Descriptive statistics for the sample of N = 28 transitional-aged youth (TAY).**

| | TAY (N = 28) | |
| --- | --- | --- |
| | n | % |
| Gender | | |
| Male | 10 | 35.7 |
| Female | 5 | 17.9 |
| Transgender male | 3 | 10.7 |
| Transgender female | 3 | 10.7 |
| Non-binary | 6 | 21.4 |
| Genderfluid/gender nonconforming exclusively | 1 | 3.6 |
| Age, in years | | |
| 18-20 | 15 | 53.6 |
| 21-23 | 8 | 28.6 |
| 24-27 | 5 | 17.9 |
| Mean, SD | 21 | 2.35 |
| Race/Ethnicity[1] | | |
| White | 11 | 42.3 |
| Middle Eastern | 2 | 7.7 |
| Hispanic | 2 | 7.7 |
| Black | 4 | 15.4 |
| Mixed race (2 or more) | 7 | 26.9 |
| Foster Care | | |
| Yes | 10 | 35.7 |
| No | 18 | 64.3 |
| Score[2] | | |
| Low (≤10) | 17 | 60.7 |
| High (≥ 11) | 11 | 39.3 |
| Mean, SD | 9.68 | 2.97 |
| Resilience Score[2] | | |
| Low (≤17) | 15 | 53.6 |
| High (≥ 18) | 13 | 46.4 |
| Mean, SD | 17.43 | 5.20 |
| Housing Status | | |
| Sheltered | 14 | 50.0 |
| Streets or "in between" sites | 6 | 21.4 |
| Transitional Living Facility (TLF) | 5 | 17.9 |
| Lives long-term with friends or family | 1 | 3.6 |
| Self-paid | 2 | 7.1 |
| PHQ-9 Score[3] | | |
| No depression (0–4) | 1 | 3.7 |
| Mild depression (5–9) | 6 | 22.2 |
| Moderate depression (10–14) | 7 | 25.9 |
| Moderately severe depression (15–19) | 6 | 22.2 |
| Severe depression (20–27) | 7 | 25.9 |
| Mean, SD | 14.56 | 6.99 |

[1] n = 2 missing values for Race/Ethnicity.

[2] Adverse Childhood Experiences. Classification of low vs. high occurs at the sample median.

[3] n = 1 missing value for PHQ-9 Total Score.

**Table 2. Differences in demographic and other characteristics between Low and High ACES classifications.**

| | ACES Low (N = 17) | | ACES High (N = 11) | | P Value |
|---|---|---|---|---|---|
| | n | % | n | % | |
| Gender | | | | | 0.7782 |
| Male | 5 | 29.4 | 5 | 45.5 | |
| Female | 3 | 17.6 | 2 | 18.2 | |
| Other | 9 | 53 | 4 | 36.6 | |
| Age, in years | | | | | 0.01522 |
| Mean, Median | 19.88 | 20 | 22.18 | 22 | |
| Race/Ethnicity[1] | | | | | 1 |
| White | 7 | 41.2 | 4 | 44.4 | |
| Other | 10 | 58.8 | 5 | 55.6 | |
| Foster Care | | | | | 0.1245 |
| Yes | 4 | 23.5 | 6 | 54.5 | |
| No | 13 | 76.5 | 5 | 45.5 | |
| Resilience Score | | | | | 0.2110 |
| Mean, Median | 16.24 | 16.00 | 19.27 | 18.00 | |
| Housing Status | | | | | 0.2766 |
| Sheltered | 10 | 58.8 | 4 | 36.4 | |
| Streets or "in between" sites | 2 | 11.8 | 4 | 36.4 | |
| Other | 5 | 29.4 | 3 | 27.3 | |
| PHQ-9 Score[2] | | | | | 0.0003 |
| Mean, Median | 10.71 | 10.00 | 21.10 | 23.00 | |

[1] n = 2 missing values for Race/Ethnicity.

[2] n = 1 missing value for PHQ-9 Total Score.

Foster care was not necessarily a respite for thirty-two percent of interviewees who entered the foster system. Some interviewees who returned to their biological parents before age 18 re-experienced the traumas that they had attempted to leave behind, thus catalyzing a decision to leave home permanently. And of the forty-six percent of TAY who identified themselves as LGBT, several detailed experiences of gender discrimination that directly contributed to their homelessness:

- *When my foster family found out I identified as trans male, they kicked me out. However, I knew my rights so I got the cops to reinstate me in the house. They kicked me out again on my eighteenth birthday. (Participant 5)*

- *When I turned 18, I started hormone therapy in secret. My mom found out in April and kicked me out. (Participant 28)*

**Themes of trauma: Unstable housing history.** Housing was a prominent stressor for every TAY that came up repeatedly. Housing situations varied from the households of family members and friends to shelters, tents, park benches, and transitional living facilities (TLFs). For each participant we inquired as to the core reason why they experienced their first instance of housing instability (S1 Table). We defined three housing groups for TAY. "Sheltered" refers to individuals staying in a temporary shelter. "TLF" refers to individuals in transitory situations that are more stable (i.e., TLP, friend's house, funding own apartment). "Streets" refers to individuals living outdoors.

**Themes of resilience and protective factors: Professional goals.** When asked about life goals, ninety-three percent of interviewees had a concrete answer. Thiry-one percent of those mentioned higher education while eighteen

**Table 3. Differences in demographic and other characteristics between Low and High Resilience classifications.**

| | Resilience Low (N = 15) | | Resilience High (N = 13) | | P Value |
|---|---|---|---|---|---|
| | n | % | n | % | |
| Gender | | | | | 1 |
| Male | 5 | 33.3 | 5 | 38.5 | |
| Female | 3 | 20 | 2 | 15.4 | |
| Other | 7 | 46.7 | 6 | 46.1 | |
| Age, in years | | | | | 0.3635 |
| Mean, Median | 20.4 | 20 | 21.23 | 21 | |
| Race/Ethnicity[1] | | | | | 0.7007 |
| White | 7 | 46.7 | 4 | 36.4 | |
| Other | 8 | 53.3 | 7 | 63.6 | |
| Foster Care | | | | | 0.4328 |
| Yes | 4 | 26.7 | 6 | 46.2 | |
| No | 11 | 73.7 | 7 | 53.8 | |
| ACES Score | | | | | 0.5778 |
| Mean, Median | 9.47 | 9.00 | 9.92 | 10.00 | |
| Housing Status | | | | | 0.5178 |
| Sheltered | 8 | 53.3 | 6 | 46.2 | |
| Streets or "in between" sites | 4 | 26.7 | 2 | 15.4 | |
| Other | 3 | 20.0 | 5 | 38.5 | |
| PHQ-9 Score[2] | | | | | 0.2402 |
| Mean, Median | 16.13 | 16.00 | 12.58 | 13.00 | |

[1] n = 2 missing values for Race/Ethnicity.

[2] n = 1 missing value for PHQ-9 Total Score.

percent were currently enrolled in an academic program (i.e., GED, certificate program, or the like). This number could be an underestimate since the youth center offers a high school program and various classes.

Aspirations included nursing school, game development, running a sticker business, working in a warehouse, joining the Marines, or finishing their graduate degree. Motivations were either driven by academic interest or stable financial opportunities. Their answers sparked reflection about the purposes of higher education.

- *I want to return to college and make money any way possible. (Participant 3)*

- *I want to prove that a lot of people were wrong about me. I was told I wasn't doing things that would get me somewhere. (Participant 6)*

- *I'm going to school for an associate's degree in psychology, behavioral science, and business management. One semester left! My larger goal is to get a PsyD. (Participant 9)*

- *"The reason why I'm not gonna give up is I think I deserve more in my life than I have now. I think this thought can help me to keep going." (Participant 1)*

The interviewees who replied with no professional goals expressed that their priorities lay in survival.

- *I'm just going to live day to day and if I die, I die. (Participant 5)*

- *I just want stable housing and more friends. (Participant 16)*

**Themes of resilience and protective factors: Close relationships.** Eight-nine percent of interviewees said they had close relationships with at least one person including parents, friends or romantic partners. Two of the three interviewees who denied having close relationships were in the low resilience category, and neither of those individuals had professional mental health support as well.

LGBT youth in our study reported slightly higher resilience scores than non-LGBT youth despite having higher ACES scores. Many described a "chosen family" met through the shelter system and others listed partners as solid sources of support. Several discussed how their access to gender-affirming care improved their mental health, increased their self-confidence, and made them feel seen and accepted in their bodies.

- *I've met friends through [shelter] and [shelter]; my partner is also a trans woman. (Participant 28)*

- *I have friends in [city]. But I feel like I only have 1–3 people here that I can talk to, and I don't feel very close to them. (Participant 17) I talk to my mom but I feel like she doesn't really understand me. I have a small social network; I like being on my own. (Participant 16)*

**Themes of resilience and protective factors: Coping strategies.** We asked interviewees how they coped with hardship and what moved them forward. Some named hobbies and social connections they turned to for comfort. Others talked about distracting themselves with risky behaviors including substance use, sex, or sleep.

- *Acrobatics, fighting... In the past I felt like I was protecting others but I wasn't protected myself. (Participant 7)*

- *I like to draw and listen to music. I also talk to my mom regularly. (Participant 16)*

- *I work a lot to distract myself. (Participant 17)*

- *"Nicotine and dick. When I get depressed and stressed, I download Tinder again." (Participant 15)*

Three quarters of our sample were currently seeing a mental health professional – a therapist, caseworker, psychiatrist, or physician – to help them manage past and ongoing traumas. Others reported they would like to see a specialist but were on a waiting list. The interviewee below described how childhood trauma directly impacted their mental health as adults.

- *After my grandma died, I developed a split personality. My mind did the only thing it could do to protect itself at that age... [it said] let's put all the strongest parts of him together and make it into a persona that protects the other part of him. We'll put this persona out onto the world to face all the pain and all the things that can possibly hurt him. When he's safe, he can back off and let the real him take the reins. (Participant 3)*

Of those who had learned coping strategies and practices for healthy navigation of traumas from a professional, the lessons were apparent.

- *"Showering when I realize I haven't been taking care of my body." (Participant 9)*

- *"I talk with staff at [shelter]. I text the national self-harm hotline. I practice deep breathing, conjuring 5 visuals." (Participant 8)*

- *"If something really bad happens, I remind myself, 'I know this feels big right now and these emotions feel big right now, but how is it going to feel in a few years? Is it even going to matter?' Or it's like, what can I do about it. I hate feeling powerless. I sometimes reach out to an old counselor who reminds me of the 7 steps to taking care of oneself. I never remember what they are until he reminds me." (Participant 4)*

## Discussion

Our study aimed to explore themes in adversity, resilience, and coping strategies among young adults experiencing homelessness, a population highly vulnerable to poorer outcomes in physical and mental well-being. These findings align with broader homelessness research while revealing population-specific nuances.

## Trauma profiles

Family violence and abuse emerged as the central thread connecting the trauma experiences across our sample. These events systematically dismantled family units that were often already vulnerable and poorly functioning. When biological families dissolved, participants faced cascading consequences: fractured relationships, housing instability, threats to personal safety, educational disruptions, and severe financial stress.

The pathway to homelessness for these youth typically involved being shuffled between multiple living arrangements. Some were forcibly removed from one location and compelled to find another, while others chose to leave when their current housing became unsafe or unstable. Having grown up without consistent support and nurturing, these youth entered young adulthood poorly equipped to navigate the complex web of personal, interpersonal, and structural challenges they encountered.

Participants who had experienced foster care or group homes reported the greatest instability and weakest support networks. They frequently described how seemingly minor rule violations—arriving ten minutes past curfew or smoking outside a shelter—resulted in immediate termination of their housing arrangements. This punitive approach left many with nowhere to turn, accelerating their path to street homelessness. This echoes concerning findings from Los Angeles cohort studies, where significant numbers of transitional-age youth identified punitive group home policies as direct catalysts for their transition to street homelessness [23]. This pattern stands in stark contrast to European approaches that prioritize harm reduction over rigid rule enforcement in youth shelter systems, suggesting alternative models that might better serve this vulnerable population [24].

Our findings align with broader patterns documented in homeless populations. The prevalence of family violence and abuse in our sample mirrors research on adult homelessness, where 62% report childhood adversity as a precursor to housing instability [25]. However, our participants' experiences reveal a key distinction from studies of homeless families, where parental bonds typically endure as protective factors even amid economic hardship [26]. For our participants, these fundamental family connections had been severed entirely.

## Resilience challenges

Understanding resilience in unhoused youth proves complex because resilience emerges from both innate characteristics, or *assets,* and environmental factors, or *resources,* in accordance with resilience theory [18]. The Brief Resilience Scale (BRS) measures resilience in terms of assets—the ability to bounce back from adversity—but this approach captures only a snapshot of an individual's stress recovery capacity. It fails to provide a comprehensive picture of how people overcome hardship across multiple life domains, build healthy relationships, and pursue meaningful goals.

This limitation became apparent in our findings. Even participants who scored in the low resilience group on the BRS demonstrated remarkable ability and determination to employ diverse coping mechanisms and displayed various adaptive characteristics, as seen in Table 4. This observation aligns with what researchers call "Windle's paradox", the reality – seen in recent adult homeless studies -- that homelessness itself often promotes resilient behaviors and complex adaptive strategies regardless of how individuals score on psychometric measures [27,28].

An intriguing pattern emerged when we examined housing status alongside resilience scores. Individuals in transitional living programs (TLPs) reported slightly higher resilience scores than those in more precarious housing situations. These data and our anecdotal observations inform our belief that stable housing creates a foundation that allows people to feel safer and more supported, knowing their basic needs are temporarily secured. This security, in turn, enables them to feel more hopeful about the future and pursue other goals such as employment, friendships, intimacy, and personal confidence—a progression that aligns with Abraham Maslow's hierarchy of needs [29].

However, the relationship between resilience and housing may be bidirectional. While stable housing may foster resilience, it's equally plausible that individuals with higher resilience are better positioned to advocate for themselves and successfully access more permanent housing arrangements. This creates a complex dynamic where resilience both influences and is influenced by housing stability.

**Table 4. Evidence-based models for promoting resilience in unhoused youth populations.**

| Key Area | Program | Comments |
|---|---|---|
| Assets: Self-Efficacy, Emotional Learning and Regulation | Career or vocation-focused workshops with opportunities to earn certificates and gain recognition for their craft | -Participation in an open art studio for the unhoused was correlated with higher life achievement including securing housing, finding employment, and selling artwork [32] |
| | Counseling and/or groupwork focused on personal strength affirmation and goal-setting | -A study found that formerly homeless individuals were more likely than currently homeless individuals to report that strength recognition, realizing that they had something to offer the world, contributed to their escape from homelessness [33] |
| | Meditative spaces in shelters and lessons in yoga, artistic reflection, and mindfulness | -Yoga interventions were shown to reduce violent behavior and increase mindfulness within a group of unhoused youth [34]<br>-Self-reflective exercises in art facilitate the circulation of messages about goals, hopes, and dreams [35] |
| | Health literacy workshops | -Higher rates of health literacy have been associated with better self-management of chronic disease [36] |
| Resources: Supportive relationships, safe and secure living environments | Peer support groups and "big-sib little sib" mentorship models for youth experiencing homelessness | - Supportive relationships were highly reported by our High Resilience group<br>- "No support" was the highest ranked ACE For LGBT-identifying TAY in our study<br>- A study found that pairing adolescents with natural mentors, especially nonfamilial mentors, reduced internalizing symptoms and substance use among participants [37] |
| | Mediating structured family therapy between unhoused youth and estranged family members/ friends to find common emotional ground and learn methods of conflict resolution | -Divorce and emotional abuse were frequently reported ACES in our group<br>-A systematic review of youth homelessness initiatives found that RCTs utilizing family therapy have found positive effects on internalizing and externalizing behaviors, family cohesion, and substance use [38]<br>-An intervention for families facing multiple trauma/loss events (primarily targeted at military families) reported reductions in distress and increases in child pro-social behaviors and family resilience |
| | County, state, and federally-funded Housing First efforts to enroll unhoused youth in transitional living programs (TLFs) | -Safe and stable environment was reported as a key area of counter ACES<br>-In a study of 47 homeless youth, resilience occurred as a function of how long youths had been without stable housing [39] |

## Housing stability as a resilience catalyst

The relationship we observed between transitional living programs (TLPs) and higher resilience scores aligns with a growing body of evidence demonstrating that housing-first interventions enhance coping capacity across diverse populations. This connection extends beyond our specific sample—recent studies have documented similar patterns, showing that housed transitional-age youth maintain significantly better odds of sustaining employment compared to their street-dependent peers [24]. These findings suggest that stable housing creates a foundation that enables young people to engage more successfully with other life domains.

Our interpretation of these findings through Maslow's hierarchy of needs offers one lens for understanding this phenomenon. However, this hierarchical model may not capture the full complexity of recovery pathways for homeless youth. Some researchers challenge the assumption that basic needs must be fully met before social and emotional development can occur, instead arguing that social connections often serve as the primary catalyst for positive change in this population [26].

This alternative perspective suggests that meaningful relationships and social support may be equally or more important than housing stability in fostering resilience. Rather than viewing needs as strictly hierarchical, this model proposes that housing, social connection, and personal development can occur simultaneously and reinforce each other in complex ways. Resource development for homeless youth should therefore adopt as multipronged an approach as possible, fostering

support and programming in all these domains. In practice, we have observed homeless youth pursuing relationships, employment, and housing opportunities in tandem, with progress in any one area supporting advancement in others.

## Implications for policy and practice

While trauma serves as a universal pathway to homelessness, resilience strategies vary dramatically by age and context. Our participants predominantly engaged in "survival-driven resilience"—resource-sharing, mutual protection, and collaborative problem-solving with peers. This contrasts sharply with older adult populations who more frequently utilize formal community services, reflecting both developmental differences and systemic barriers that make formal services less accessible to youth [25].

These findings reveal a critical gap in service delivery: most interventions emphasize individual case management designed for adults, yet our participants tended to respond better to peer-based survival networks. Effective interventions for transitional-age youth must blend housing stability with trauma-informed peer support models, honoring the collaborative strategies youth have developed while providing stable foundations [24]. Furthermore, in accordance with Windle's paradox we should recognize that these youth have started to develop a set of self-advocacy and learned resilience skills unique to this population, and programs should seek to nurture, reinforce, and introduce these skills in a safe setting.

## Limitations

Several limitations must be considered when interpreting our findings. This mixed methods, cross-sectional study drew from a convenience sample of youth presenting to a single free clinic serving unhoused transitional-age youth. Most significantly, our data potentially excludes a substantial population of unhoused young adults who avoid formal services entirely. This population may differ systematically from our participants in important ways, including trauma exposure, coping strategies, trust in institutions, or severity of mental health challenges. Their exclusion limits our understanding of the full spectrum of experiences among unhoused youth.

The small sample size and single geographic location further constrain our findings. These factors reduced the statistical power of our analyses and limit generalizability to unhoused youth in other regions where service systems, local policies, or demographic characteristics may differ substantially. Consequently, our findings should be viewed as hypothesis-generating rather than definitive, and the associations we identified do not establish causality.

Methodological constraints also shaped our data collection. The semi-structured interview format, while allowing for rich qualitative insights, meant that participants responded to open-ended questions in highly varied ways. The clinical setting imposed time limitations that may have prevented natural discussion of certain topics—for instance, a participant's failure to mention current hobbies or interests may reflect conversational flow rather than the absence of these activities. Additionally, to meet journal word limitations, we necessarily truncated or selected portions of longer participant quotes, which may not fully capture the complexity of their experiences.

These limitations underscore the need for future research employing larger, multi-site samples and longitudinal designs that can better establish causal relationships and include harder-to-reach populations of unhoused youth who remain outside formal service systems.

## Conclusion

Our findings highlight a critical need to move beyond simply addressing trauma to actively cultivating positive experiences that can counteract its effects. Future research should explore the potential of "counter-ACES"—positive childhood experiences such as stable family relationships and high emotional support—to guide intervention development [11,30]}. These advantageous experiences have been linked to improved adult outcomes including higher productivity, stronger internal locus of control, enhanced adaptive traits, and better cardiovascular health.

Residential stability represents one of the most powerful counter-ACES interventions available. Housing First models, which prioritize safe housing for anyone experiencing homelessness—particularly those with mental health or substance use challenges—have demonstrated significant benefits [31]. These include reduced mental health symptoms, decreased criminal justice involvement, lower emergency department utilization, and improved self-reported quality of life. Our findings support this model but suggest that housing alone is insufficient; effective interventions must address the complex interplay between trauma, resilience, and peer support that characterizes this population. In Table 4, we present intervention models grounded in resilience theory that integrate trauma-informed peer support—an approach that addresses both the immediate and long-term needs of this vulnerable population.

Public health approaches to youth homelessness must involve primary prevention to target the social determinants that create pathways to homelessness such as family violence, systemic inequities, and institutional failures that impact at-risk youth. Secondary prevention efforts should focus on building resilience among youth already experiencing housing instability, recognizing that traditional adult-focused services may not adequately serve transitional-age populations.

While we cannot undo the adverse experiences our participants have endured, transitional-age youth centers can implement trauma-informed, evidence-based strategies that nurture resilience-building behaviors. The path forward requires recognizing that unhoused youth possess inherent strengths and adaptive capacities that, when properly supported, can serve as foundations for recovery. By building interventions that honor these existing resilience strategies, we can create more effective response systems for this unique population.

## Supporting information

**S1 Text.  Adverse childhood experience questionnaire for adults (Fig A), Philadelphia expanded ACES questionnaire (Fig B), PHQ-9 (Fig C), and Brief Resilience Scale (BRS) (Fig D).**
(DOCX)

**S1 Table.  Core reason for first instance of housing instability.**
(DOCX)

## Acknowledgments

We thank the staff at One Community Health and WIND Youth Services for supporting our operations within the One Community Health Teen Clinic such as the provision of private interview spaces.

**Dedication:** The authors dedicate this work to the memory of Machelle Wilson, their esteemed colleague and friend, who served tirelessly as the statistician on several of their resilience projects. Her expertise, commitment, and generous spirit were vital to every stage of this work. Machelle's sudden and tragic passing is a profound loss to their team and to the broader scientific community. She will be deeply missed, both professionally and personally.

## Author contributions

**Data curation:** Hilary Aralis.

**Investigation:** Shruti Arora, Allison Ong, Michael Wilkes.

**Methodology:** Shruti Arora, Allison Ong.

**Project administration:** Shruti Arora, Michael Wilkes.

**Supervision:** Allison Ong, Michael Wilkes.

**Visualization:** Shruti Arora, Allison Ong, Hilary Aralis.

**Writing – original draft:** Shruti Arora, Allison Ong.

**Writing – review & editing:** Shruti Arora, Allison Ong, Michael Wilkes, Andres F Sciolla.

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
