## [Decision Letter · Decision Letter 0]

5 Mar 2025

PMEN-D-24-00588

Trauma and Resiliency in Unhoused Young Adults: A Qualitative Study

PLOS Mental Health

Dear Dr. Ong,

Thank you for submitting your manuscript to PLOS Mental Health. After careful consideration, we feel that it has merit but does not fully meet PLOS Mental Health’s publication criteria as it currently stands. Therefore, we invite you to submit a revised version of the manuscript that addresses the points raised during the review process.

The manuscript has been evaluated by two reviewers, and their comments are available below.

The reviewers have raised a number of major concerns regarding the reporting in the methodology/results as well as the overall structure of the manuscript.

Could you please carefully revise the manuscript to address all comments raised?

We look forward to receiving your revised manuscript.

Kind regards,

Avanti Dey, PhD

Staff Editor

PLOS Mental Health

Journal Requirements:

For more information about figure files please see our guidelines: https://journals.plos.org/mentalhealth/s/figures

https://journals.plos.org/mentalhealth/s/figures#loc-file-requirements

2. In the online submission form, you indicated that your data will be submitted to a repository upon acceptance. We strongly recommend all authors deposit their data before acceptance, as the process can be lengthy and hold up publication timelines. Please note that, though access restrictions are acceptable now, your entire minimal dataset will need to be made freely accessible if your manuscript is accepted for publication. This policy applies to all data except where public deposition would breach compliance with the protocol approved by your research ethics board. If you are unable to adhere to our open data policy, please kindly revise your statement to explain your reasoning and we will seek the editor's input on an exemption.

Additional Editor Comments (if provided):

Reviewers' comments:

Reviewer's Responses to Questions

**Comments to the Author**

1. Does this manuscript meet PLOS Mental Health’s publication criteria?

Reviewer #1: Partly

Reviewer #2: No

2. Has the statistical analysis been performed appropriately and rigorously?

Reviewer #1: No

Reviewer #2: No

3. Have the authors made all data underlying the findings in their manuscript fully available (please refer to the Data Availability Statement at the start of the manuscript PDF file)?

Reviewer #1: No

Reviewer #2: No

4. Is the manuscript presented in an intelligible fashion and written in standard English?

Reviewer #1: Yes

Reviewer #2: No

Reviewer #1: This article addresses the important issue of the relationship between adverse childhood experiences and mental health problems in youth experiencing of homelessness, while also taking into account resilience. These are very interesting and important studies that should be the basis for further research and the definition of clear health policies, as well as prevention and intervention. However, I have identified some issues that should be addressed in a re-release.

1. The authors write on page 6 (lines 118-119) that they used "quantitative statistics including chi-square, MEANS procedure, NPAR1WAY procedure, and the Kruskal-Wallis test." However, they do not report any statistics. I believe that statistics should be presented to improve strenght and quality of this manuscript. Authors can compare ACE participants in terms of depression, generalized anxiety, and resilience. Since all of these variables are continuous, Mann-Whitney U-test should be used for this purpose. The subjects should be divided into 2 groups - with higher and lower ACE experience, using a cut-off of 4 (based on the literature), or based on the mean or median. In addition, the association between categories of ACEs, depression, generalized anxiety, and psychological resilience should be compared with demographic variables (including gender, age, race, and housing situation) using the Pearson Chi-square test of independence.

2. I suggest to add more data to Table 1, including ACE and resilience categories.

3. Always report the mean, median, and standard deviation as basic subject characteristics in the results section.

4. Methods and appendix 1 does not include GAD-7, while it is included in Table 1. Please add the missing information.

5. Each abbreviation in the figures should be explained in a note below the figure (e.g., ACES, TLF).

6. It would be worthwhile to add an Excel file with the raw data, as reccomended by the journal, to the supplementary materials.

Reviewer #2: The manuscript provides a valuable exploration of childhood adversity, resilience, and coping strategies among young adults experiencing homelessness. It highlights critical social and psychological challenges, presents insightful interpretations, and acknowledges key methodological limitations. However, in its current form, the manuscript requires substantial revisions to meet the standards for publication. Improvements in clarity, depth of analysis, organisation, and scholarly rigour are essential to enhance its overall impact and contribution to the field.

Introduction

1. The use of a direct quote from a study participant is engaging and humanises the topic. However, it is not immediately clear how this quote connects to the broader discussion on youth homelessness, adversity, and resiliency. Consider briefly introducing the relevance of the quote before presenting it to help the reader understand its significance.

2. The transition between topics (youth homelessness, ACEs, resiliency, and the study’s purpose) could be smoother.

For example, the paragraph on ACEs jumps from childhood trauma to epigenetic theories of disease transmission, which, while relevant, feels abrupt. A clearer bridge between ACEs and resiliency would improve coherence.

3. The definitions of ACEs and resiliency are well-integrated, but some sentences are overly detailed or redundant.

Example: "ACES also impact the youth and adolescent years with links to developmental and educational delays, obesity, asthma, sleep issues, frequent infection, and higher levels of anxiety and depression." The list is extensive and somewhat repetitive given the earlier mention of poor physical and mental health outcomes.

4. The paragraph about the research team's involvement in a youth center is valuable but could more clearly justify why this study is necessary beyond personal experience. Consider specifying gaps in existing research or explaining why a qualitative approach is particularly useful in this context.

5. The final paragraph presents the study’s aim well but could be more impactful. Instead of saying the study "generates hypotheses for care advancement," specify how findings could inform policy, intervention programs, or clinical practice.

Methods

1. While verbal consent is noted, the reason for using verbal rather than written consent is not explained.

2. The recruitment process only includes TAY checking into the clinic, which may exclude unhoused youth who do not seek clinic services. Acknowledge this limitation and discuss how it might affect generalisability.

3. The study mentions the use of both qualitative and quantitative analyses; however, the methodological details for both are insufficient. Please provide a clearer description of the thematic analysis process, including coding strategies, how themes were identified, and steps taken to ensure rigour (e.g., interrater reliability). Additionally, elaborate on the quantitative analysis by specifying the statistical tests used, the rationale behind their selection, and how they were applied to the data.

4. It is unclear how participants were categorised into low, medium, and high resiliency (e.g., cutoff scores).

5. The final sentence repeats information about IRB approval and verbal consent unnecessarily.

Results

1. The study mentions associations between housing status, PHQ-9, and resiliency scores, but does not report statistical significance. If statistical tests (e.g., chi-square, Kruskal-Wallis) were conducted, report p-values and effect sizes to support claims of relationships. If findings are exploratory, explicitly state that no statistically significant relationships were found.

2. The study divides participants into low, medium, and high resiliency but does not explain: Why those score ranges were selected (e.g., based on prior research or distribution of scores?). Whether these categories were validated in previous studies.

3. The qualitative themes are well-structured, but the coding process is not referenced in this section. Briefly mention how themes were identified (e.g., inductive vs. deductive coding, number of coders, intercoder agreement).

Discussion

1. While the discussion is descriptive, it lacks comparative engagement with prior research. Incorporate references to other studies on homelessness, resilience, or trauma to contextualize findings. How do these findings compare with resilience levels in other homeless populations (e.g., adults, families)? Are similar trends observed in other geographic regions or demographic groups?

2. The paragraph states, “Innate qualities such as resiliency are challenging to quantify.” This implies that resilience is entirely innate, which is debatable, as resilience is also shaped by environmental and social factors. Reframe this to acknowledge both inherent and learned components of resilience.

3. The transition between topics (e.g., from trauma to resilience, and then to limitations) is somewhat abrupt. Introduce a clearer transition sentence between major sections. Consider structuring the discussion into subheadings (e.g., Key Findings, Interpretation, Implications, Limitations).

4. The discussion touches on the role of TLPs in fostering resilience but does not elaborate on practical recommendations. Expand on policy implications by discussing: How can TLPs be improved to better support youth?

What interventions could help increase resilience among those with lower scores?

5. The text references Figure 3 but does not describe it in detail. Provide a brief explanation of what Figure 3 illustrates to ensure clarity for readers.

**Do you want your identity to be public for this peer review?** For information about this choice, including consent withdrawal, please see our Privacy Policy

Reviewer #1: No

Reviewer #2: **Yes: ** Suen Yi Nam

---

## [Editor Report · Decision Letter 1]

11 Jul 2025

Trauma and Resilience in Unhoused Young Adults: A Mixed Methods Study

PMEN-D-24-00588R1

Dear Dr. Ong,

We are pleased to inform you that your manuscript 'Trauma and Resilience in Unhoused Young Adults: A Mixed Methods Study' has been provisionally accepted for publication in PLOS Mental Health.

Best regards,

Karli Montague-Cardoso

Staff Editor

PLOS Mental Health